# The Use of Cerium Compounds as Antimicrobials for Biomedical Applications

**DOI:** 10.3390/molecules27092678

**Published:** 2022-04-21

**Authors:** Emilia Barker, Joanna Shepherd, Ilida Ortega Asencio

**Affiliations:** Mechanisms of Health & Disease, School of Clinical Dentistry, The University of Sheffield, Sheffield S10 2TA, UK; emilia.barker@sheffield.ac.uk (E.B.); j.shepherd@sheffield.ac.uk (J.S.)

**Keywords:** cerium nitrate, nanoceria, antimicrobial, wound healing

## Abstract

Cerium and its derivatives have been used as remedies for wounds since the early 20th century. Cerium nitrate has attracted most attention in the treatment of deep burns, followed later by reports of its antimicrobial properties. Its ability to mimic and replace calcium is presumed to be a major mechanism of its beneficial action. However, despite some encouraging results, the overall data are somewhat confusing with seemingly the same compounds yielding opposing results. Despite this, cerium nitrate is currently used in wound treatment in combination with silver sulfadiazine as Flammacérium. Cerium oxide, especially in nanoparticle form (Nanoceria), has lately captured much interest due to its antibacterial properties mediated via oxidative stress, leading to an increase of published reports. The properties of Nanoceria depend on the synthesis method, their shape and size. Recently, the green synthesis route has gained a lot of interest as an alternative environmentally friendly method, resulting in production of effective antimicrobial and antifungal nanoparticles. Unfortunately, as is the case with antibiotics, emerging bacterial resistance against cerium-derived nanoparticles is a growing concern, especially in the case of bacterial biofilm. However, diverse strategies resulting from better understanding of the biology of cerium are promising. The aim of this paper is to present the progress to date in the use of cerium compounds as antimicrobials in clinical applications (in particular wound healing) and to provide an overview of the mechanisms of action of cerium at both the cellular and molecular level.

## 1. Introduction

Cerium was discovered in 1803 by Berzelius and Hisinger and is a member of the lanthanides family (or rare earth materials); the most studied lanthanides are lanthanium (Ln), cerium (Ce), neodymium (Nd), gadolinium (Gd), ytterbium (Yb) and yttrium (Y). Despite their name, rare-earth elements are—with exception of the radioactive protheium—relatively plentiful in the Earth’s crust, with cerium being the 25th most abundant element (more abundant than copper). Lanthanides are found in a variety of minerals such as bastnaesite and monazite, which are extracted commercially for use in the glass industry, flint manufacture and radiation shielding.

As a group, the lanthanides have no physiological role and are unable to penetrate intact mammalian cell membranes but can enter the cytoplasm of effete cells [1]. Unlike the other lanthanide series, which usually exhibit a trivalent (+3) state, the cerium atom can exist in either the +3 (fully reduced) or +4 (fully oxidized) state, because it has two partially filled subshells of electrons, 4f and 5d, with several excited substrates predicted [2]. The relative amount of cerium ions Ce^3+^ Ce^4+^ is a function of particle size. In general, the fraction of Ce^3+^ ions in the particles increases with decreasing particle size [3]. Cerium forms a variety of salts, such as cerium nitrate (as a colourless, aqueous hexahydrate [Ce(NO_3_)_3_·6H_2_O] solution), which attracts much medical interest. Aqueous solutions of Ce^4+^ salts (without complexation) are stable only in highly acidic solutions (pH < 3). Ce^4+^ is a powerful oxidant without biological relevance [4]. On the other hand, Ce^3+^ is very resistant to oxidation and reacts only with very strong oxidants. In the case of cerium oxide nanoparticles, these are remarkable in that they can have a dual role as an oxidation catalyst and reduction catalyst, depending on the reaction conditions. These activities derive from the expedient mutation of the oxidation state between Ce^4+^ and Ce^3+^. The cerium atom has the ability to easily and drastically adjust its electronic configuration to best fit its immediate environment [5]. It also exhibits oxygen vacancies, or defects, in the lattice structure that arise through loss of oxygen and/or its electrons, alternating between CeO_2_ and CeO_2-x_ during redox reactions. The Ce^4+^ and the low formation energies of surface vacancies are important for oxidation, whereas the Ce^3+^ and electron shuffling within the lattice oxygen vacancies provide power for reduction. Moreover, the addition or removal of oxygen atoms in the oxidizing or reducing process involves a minimal reorganization of the skeleton arrangement of the cerium atoms and the retention of the fluorite structure [5]. This structural property facilitates the regenerative ability of cerium oxide nanoparticles to the initial state, and thereby can be recycled to act catalytically.

Nanoparticles of cerium oxide containing Ce^3+^ are of particular interest as potential therapeutics to provide a persistent antioxidant effect in the treatment of a number of medical conditions. They are used therapeutically in several biomedical applications, mostly for ROS related diseases including cardiac diseases, Alzheimer’s disease, and cancer [6].

Trivalent Ce exhibits similar chemical properties to calcium. Calcium in its bivalent state is very similar to trivalent cerium in terms of size, bonding and preferences to donor atoms. This similarity is reflected in their natural occurrence, where cerium is always found together with calcium (e.g., in apatite). In biological systems, they have a tendency to precipitate together, hence the high affinity of Ce to the mineral bone matrix [7] and in induction of local soft-tissue calcification [8]. The mechanism of this calcergic action is not fully understood, but it has been hypothesized that Ce precipitates with pyrophosphate, resulting in the loss of the calcification-inhibitory function of pyrophosphate, and these precipitates form crystallisation nuclei on which calcium and pyrophosphate accumulate to form apatite [9].

Because of their similar ionic radius (1.00 Å for Calcium^2+^ vs. 1.01 Å for Cerium^3+^), lanthanide ions are able to replace calcium in many biomolecules, without necessarily replacing functionality. For example, they interfere in calcium-dependent reactions of the blood clotting cascade, such as activation of prothrombin [10] and factor XIII [11]. These anticoagulant properties found clinical application as antithrombotic drugs prior to the widespread availability of heparin [12]. Ce^3+^ is capable of binding to the Ca^2+^/Mg^2+^-ATPase of the sarcoplasmic reticulum of skeletal muscle [13] and of inhibiting the active transport of calcium through mitochondrial membranes [14,15]. Moreover, Ce^3+^ is a potent blocker of neuronal low voltage–activated (T-type) calcium channels [16,17], high voltage–activated calcium channels of presynaptic nerve terminals [18], and skeletal muscle cells [19]. Moreover, A-type potassium channels of adrenal cortical cells are inhibited by binding Ce^3+^ to sites that are not specific for calcium [20], but currents through type A γ-aminobutyric acid (GABA)–activated chloride channels of rat dorsal root ganglion neurons are enhanced [21]. Ce^3+^ is also capable of binding to calcium-binding sites of the N-terminal domain of calmodulin (CaM), which mediates intracellular responses to calcium fluxes, in a cooperative manner [22] and of substituting for calcium in the regulation of calcium/CaM-dependent enzymes such as phosphorylase kinase [23]. Johansson et al. reported that the strong interaction between these two ions is supported by findings that when rare earth metals were administered (for example, CeCl_3_ and HoCl_3_), they observed two reactions: first, an increase of serum calcium and phosphorus induced by small doses of metallic chlorides, and second, a decrease of both elements after injection of high amounts [24]. The mechanism of this response is unknown, but it could be assumed that lanthanides do not follow a simple dose-response relationship but may turn from enhancing to inhibiting, depending on the dose [25].

The phenomenon of antimicrobial resistance (AMR) has emerged among pathogenic bacteria since the beginning of the antibiotic era as a consequence of the selective pressure generated by the extensive use of antibiotics in human and veterinary medicine. Nowadays, pathogenic organisms expressing Multi Drug Resistance (MDR) phenotypes are among the most important causes of infections in hospital and community settings. Therapeutic options for infections originated by MDR pathogens are limited and often ineffective; therefore, new drugs are urgently needed [26,27,28,29,30]. To overcome AMR and improve antibiotic stewardship, one strategy would be the use of combination drugs or the development of adjuvants that, acting jointly with licenced agents, can enhance their antimicrobial activity against resistant strains [31]. Development of antibiotic adjuvants does not necessarily imply the discovery of new targets in bacterial cells; on the contrary, most known targets can be exploited (e.g., beta lactamase inhibitors), or even non-specific compounds such as an outer membrane permeabiliser can be used to increase the activity of antibiotics [32]. The outer membrane of Gram-negative bacteria is a semipermeable barrier that has inherent resistance to most antibiotics [33]. Hydrophilic antibiotics such as beta-lactams utilise channel-forming proteins, known as porins, as an entry point. However, the uptake efficiency is limited as these proteins represent <1% of the surface area and, additionally, the rate of uptake into the cell is often reduced by the presence of efflux pumps. The passage of hydrophobic components can be further prevented by LPS (polyanionic lipopolysaccharide), which is present in the outer membrane, and which is stabilised by divalent cations. The displacement of the stabilising cations, by polycations such as polymaxins, aminoglycosides or cationic peptides [33] makes the outer membrane increasingly permeable to other compounds [34].

This review begins with a brief introduction of the antibacterial mechanism of cerium nanoparticles, particularly focusing on cerium nitrate and cerium oxide, followed by a summary of the antibacterial applications of cerium with respect to wound healing and future outlooks in development of cerium-based nanoparticles as new antibiotics/antifungals.

## 2. The Antimicrobial Properties of Cerium-Based Compounds

The bacteriostatic (prevention of bacterial growth) properties of Ce were first recognised at the end of 19th century [35] and paved the way for its incorporation into topical antiseptics in human and veterinary medicine. These mixtures, ranging from Ce^3+^ acetate solutions, as well as powders and ointments containing cerium (III) stearate, were successfully used for the treatment of wounds, which included burns, weeping eczema, intertrigo and decubitus, skin gangrene, impetigo contagiosa and other skin diseases [36]. Ce^4+^ potassium sulphate was applied as an antiseptic powder as ‘ceriform’, based on the idea that the bacteriostatic effect of Ce^4+^ is a result of its oxidising properties [36].

In 1947, Burkes and McCleskey carried out a systematic investigation of the bacteriostatic properties of cerium (cerium^3+^ chloride, cerium^3+^ nitrate, and cerium^4+^ sulphate), Lanthanum and the non-rare element thallium were tested against a panel of 39 bacterial species across 16 genera, including Gram positive *Staphylococcus aureus* and Gram-negative *Pseudomonas aeruginosa.* They found that cerium nitrate was an effective bacteriostatic agent against the whole spectrum of bacteria, but was pH-dependent, with its greatest effect at slightly acidic pH values. The most susceptible genera were the pseudomonas, with inhibition of growth at concentrations 0.001–0.004 M of cerium nitrate, *Escherichia* and *Salmonella* species at 0.005 M, while the effect on *S. aureus* was observed at almost twice that concentration. As a comparison, subsequent tests using cerium chloride against 35 common fungi failed to achieve any fungal inhibition at all, even at concentrations higher than those which previously resulted in bacterial growth inhibition [37]. A further study using chlorides of scandium, yttrium, lanthanum, cerium, praseodymium, neodymium, samarium, europium and ytterbium showed that Gram-negative bacteria are generally more susceptible than Gram positive bacteria [38]. In 1968, Sobek and Talburt studied the effects of cerium nitrate on *Escherichia coli*. They found ready uptake of cerium into the cell cytoplasm, inhibition of cellular respiration, oxygen uptake and glucose metabolism. The cell wall remained intact; however ‘knob-like protrusions’ were visible, similar to those observed on fungi, which suggest a disruption of the cell wall [39]. In comparison, no such effects were observed in mammalian cells. These findings contributed to clinical implementation of topical cerium(III) nitrate (applied as a cream or in saline solution) in the treatment of extensive, life-threatening burns, which resulted in a nearly 50% reduction in the death rate compared to the anticipated mortality if the patient had been treated with silver nitrate [40].

Bacterial flora recovered from treated wounds tend to be dominated by Gram-positive bacteria; therefore, a combination with the complementary acting silver sulfadiazine (cerium nitrate/silver sulfadiazine (CN-SSD) was recommended in the 1970s [40]. Silver sulfadiazine (SSD) was introduced in 1968 [41] by substitution of a single hydrogen ion of sulfadiazine by silver. It was an attempt to replicate the known antimicrobial properties of silver nitrate in burns patients without the problems previously encountered of electrolyte disturbances, toxicity and staining of burn and bedding [42]. Its widespread activity against a variety of microbes including Staphylococci, Pseudomonas and fungi soon made it a preferable agent for topical application [43]. Death from sepsis in patients surviving the original hemodynamic and pulmonary insults from near-total burns could be effectively prevented with use of this combination, and halving of the mortality rate compared to prediction [44]. The increased effectiveness of SSD in comparison to sulfadiazine was attributed to the incorporation of the dissociated silver ion into the bacterial nucleotide sequences with inhibition of replication [45].

Several in vitro studies [40,46,47,48] against common burn unit pathogens aimed to assess antimicrobial activity of SSD and CN-SSD by measuring the diameter of the induced zone of inhibition, a common test for the efficacy of putative antimicrobials, yielded conflicting results, which could be explained by lack of reliable testing methods. Ce easily binds to protein and phosphates, which induces precipitation in many liquid test media, reducing the concentration of active components and potentially affecting the results. Similarly, silver sulfadiazine is poorly soluble and can produce poorer than anticipated results in agar disc diffusion tests with the zone of inhibition as the output [48]. Although results of in vitro studies concerning the synergism or antagonism of Ce have been conflicting, and the first prospective, randomised studies comparing silver sulfadiazine plus cerium nitrate to silver sulfadiazine alone failed to demonstrate any advantage of the combination in adults [49] and children [50], a more recent trial showed the greater efficacy of the combination therapy in terms of faster re-epithelialization of skin, earlier readiness for autologous skin grafting, reduced duration of hospitalisation and potential reduced mortality [51]. Marone et al. studied the antimicrobial activity of SSD, alone and in combination with cerium nitrate (CN), gentamicin and amikacin against 130 recent clinical isolates, including multiresistant bacteria such as methicillin-resistant *Staphylococcus aureus* (MRSA) or *Pseudomonas aeruginosa* and showed that the combination of SSD and CN was as active as SSD alone [52]. This is a significant finding for clinical use in that any differences in efficacy of several commercially available silver dressings can be potentially overcome by the addition of CN to the treatment regimen.

The timeline of discovery of cerium and its antibacterial properties is presented in Table 1.

As mentioned earlier, cerium oxides are also compounds of interest and have been studied due to their antibacterial properties. In the next section, we present the structure of cerium oxide nanoparticles and their redox properties to explain how they influence their antibacterial properties on Gram-positive and Gram-negative bacteria. This provides an insight into the mechanism of Cerium Oxide nanoparticles as antibacterial agents and helps open up perspectives on their applications in biomedical areas in the future.

## 3. Antimicrobial Activity of Cerium Oxide Nanoparticles

In recent years, the interest in positively-charged metal oxide nanoparticles (MeO-NPs) as potential antimicrobial agents against drug-resistant pathogens [85] has increased enormously. The nanometre size of metal oxide NPs is strictly related to their antimicrobial activity as well as their physical and chemical properties [86]. Specifically, ceria nanoparticles (NPs) have been extensively studied for a variety of potential applications in several fields, including nanomedicine [87].

Cerium oxide can exist as both CeO_2_ and Ce_2_O_3_ in the bulk state [88] and shows catalytic activity due to the redox behaviour of cerium [89]. It can adopt a fluorite crystalline lattice structure, due to which it has a highly reactive surface area for the neutralisation of free radicals [90]. With the decrease in the size of nanoceria, oxygen vacancies are formed in their lattice structure [91] creating oxygen defects, thereby acting as a free radical scavenger in the physiological environment [92].

The ability of nanoceria to act in an antioxidant capacity has been well established [93], but investigations into their antimicrobial properties are still undergoing. Recently it has been demonstrated that coated ceria NPs are able to inhibit the growth of *P. aeruginosa* by up to 50% [94]. Thill et al. [95] demonstrated the cytotoxicity of CeO_2_ NPs against *E. coli,* and Shah et al. demonstrated that dextran coated cerium oxide nanoparticles are able to induce toxicity against *E. coli* [67]. Pelletier et al. showed that cerium oxide NPs exert moderate bactericidal activity *E. coli* and *Bacillus subtilis* [66]. Moreover, several studies have demonstrated that the morphology, size and composition of CeO_2_ NPs surface, characterise their antibacterial properties, as is the case with other MeO-NPs [96,97,98]. In general, Ce nanoparticles (CeNPs) are antimicrobial against both Gram-positive and Gram-negative bacteria, with the greatest activity observed against Gram-negative bacteria (*E. coli*). This could be due to Gram-positive bacteria having a thick outer layer of peptidoglycan that contains linear polysaccharide chains with short peptides that together form a rigid structure which is difficult to penetrate with Ce NPs. However, some authors reported the opposite findings. For example, Gopinath et al. [99] investigated CeO_2_ NPs against Gram-positive and Gram-negative bacteria and found that for very high concentrations (10 mg/mL) the inhibition zone was more pronounced in Gram-positive bacteria. The observed results could be attributed to a binding of metal and metal oxide nanoparticles onto the bacterial cell wall due to the electrostatic attraction between the negatively charged bacteria and the positively charged nanoparticles. This is not a new finding. Thill and other authors [66,95,100] reported earlier that CeO_2_ NPs adsorb via electrostatic attraction to the bacterial surfaces but do not penetrate them. The strong electrostatic interactions between NPs and the membrane results in nanoceria adherence to the membranes for extended periods, allowing Ce^4+^ atoms close to the membrane surface to be reduced to Ce^3+^, resulting in oxidative stress on the major components of the membrane such as lipids and/or proteins [95]. The oxidation of the bacterial cell would create mesosoma-like structures, therefore several elementary and essential functions, such as DNA replication and cell division, are changed and, consequently, the surface area of the bacterial cell membrane is increased due to formation of membrane invaginations [72].

Thill et al. [95] suggested three types of interactions between bacteria and CeNP: (1) direct contact, or adsorption, (2) oxi-reduction, and (3) toxicity.

In direct contact, nanoceria is directly adsorbed on the bacterial cell and damages the outer cell wall, which further leads to the generation of intracellular ROS [101]. Armugam et al. [72] demonstrated that positively charged nanoceria are well adsorbed on the bacterial cell due to electrostatic interaction. Due to this interaction, nanoceria interfere with the bacteria cell membrane and bind with the mesosome. This causes perturbance in the mesosomal functions of cellular respiration, DNA replication, and cell division. In addition, some NPs with uneven surface textures contribute to the mechanical damage of the cell membranes of *E. coli* [102], which can explain some reports of higher susceptibility of Gram-positive bacteria to nanoceria.

Oxi-reduction takes place when modifications occur on the surface of the nanoparticle and the bacteria. The Ce^4+^ charge of the nanoparticles is reduced to Ce^3+^ in the presence of the bacteria (*E. coli*), resulting in oxidative stress on lipids and/or proteins in the plasma membrane of the microorganism, or through cellular metabolism electron uptake. It is important to highlight that no reduction of Ce^4+^ was observed in abiotic culture medium [95,103]. Toxicity involves the impairment of cellular respiration, as observed by differences in gene expression, in nanoparticulate exposed and unexposed *E. coli.* The low level of succinate dehydrogenase and cytochrome b terminal oxidase gene expression in the experimental group indicates that cerium attacks electron flow and bacterial respiration [66]. With respect to the fungal species *Candida albicans*, it is believed that the interaction between cerium and components of the fungal cell wall can cause irreversible changes, such as blocking fungal enzymatic activity [104].

The third mechanism is mediated by oxidative stress, induced by the generation of reactive oxygen species (ROS) in vivo as a result of reversible conversion between Ce^3+^ and Ce^4+^ on the surface of bacterial membranes [105]. ROS can attack the nucleic acids, proteins, polysaccharides, lipids and other biological molecules resulting in loss of their function, eventually killing and decomposing bacteria [106]. Although CeO_2_ can be excited to produce ROS by ultraviolet (UV) irradiation, there are very few studies concerning bacterial activity by using CeO_2_ alone. Usually, CeO_2_ is combined with other photocatalysts such as TiO_2_. In the presence of CeO_2_, the band gap can be changed in the host lattice of photocatalysts, which improves the photocatalytic activity of TiO_2_ [107].

Finally, Ce (IV) ions could induce hydrolysis of a DNA oligomer into fragments, which then could be successfully manipulated by natural enzymes [108]. Extracellular DNA (eDNA) is one of the important components in the process of biofilm formation, making biofilms hard to eliminate. Therefore, taking advantage of Ce-based nanozymes with deoxyribonuclease (DNase) mimetic activity could lead to high cleavage ability toward eDNA and disrupt the established biofilm [109,110].

Bellio et al. found that CeO2 NPs with a diameter of ~10 nm are able to induce a slight permeabilization of biological membranes in a dose-response manner, without any evident damage [111]. They also demonstrated that their CeO_2_ particles have a core-shell nanostructure in relatively large particles; the core has a composition close to stoichiometric CeO_2_ (Ce^4+^) and the surface is close to Ce_2_O_3_ (Ce^3+^). As previously reported [112], semi quantitative analysis of the Ce 3d core-level peak revealed a concentration of Ce^3+^ ions of 24.7%. Therefore, they proposed that the composition of NPs can be schematized as a crystalline CeO_2_ (Ce^4+^) core part and an amorphous Ce_2_O_3_ (Ce^3+^) part at the surface of the nanoparticle. This might be responsible for the high biological activity of nanocrystalline cerium dioxide, based on binding of reactive oxygen compounds and radicals deleterious for living systems [87]. Moreover, the positively charged cerium oxide NPs might displace the divalent cations that stabilise the lipopolysaccharide of the outer membrane, increasing the permeability of the outer membrane to both hydrophobic and hydrophilic substances, as also previously demonstrated for other polycationic molecules [34]. The overall result is that the surface area permeable to antibiotics, which is physiologically restricted to porins, is increased, allowing passive diffusion following the concentration gradient. The consequence is a net movement of antibiotics from the area of high concentration, the environment, to the area with lower concentration, the periplasmic space.

Another relevant factor in antimicrobial activity is altering of nanoparticle surface charge by the culture medium pH. For example, the medium pH can alter nanoparticle surface charge and thus adsorption affinity of the particles toward the bacteria [66]. Other factors are size [113], concentration [72], surface coating [114] and surface chemistry [115]. Recently, Mishra et al. found that addition of some elements to cerium oxide also showed some antibacterial activity [116]. CeVO_4_ nanoparticles showed excellent antibacterial activity against *Streptococcus mutants* and *Streptococcus pyogenes* with minimum inhibitory concentration (MIC) values at 200 µg/mL and against *Vibrio cholera*, *Salmonella typhi* and *Shigella flexneri* with MIC values at 350 µg/mL. Several researchers have studied the antibacterial activity of nanoceria against different strains of bacteria, as summarised in Table 2.

CeO_2_ microspheres produced by green synthesis showed significant antibacterial activity against *Escherichia coli* and *Staphylococcus aureus* with ZOI (zone of inhibition, a qualitative method used clinically to measure antibiotic resistance and industrially to test the ability of solids and textiles to inhibit microbial growth) values of 4.67 and 3.33 mm, respectively. The ZOI is dependent on the type of bacteria, the concentration, surface area, shape and size of nanoparticles [128]. Patil et al. [129] used cerium nitrate, pectin and ammonia solution as precursors to synthesise nanoceria. The synthesised nanoparticles were spherical, with an average particle size of ≤40 nm, showed antioxidant and antibacterial properties, and were proven non-toxic toward living tissues. Nanoparticles synthesised using a fungus-mediated approach (Mycosynthesis) were small in size (5–20 nm) with spherical shape and possessed antibacterial activity [130]. Arumugam et al. [72] used a plant-mediated approach to synthesise nanoceria by using *Gloriosa superba* leaf extract and cerium chloride salt as a precursor. The synthesised nanoceria were spherical with particle size of 5 nm and showed antibacterial properties.

## 4. Clinical Applications of Cerium Compounds as Antimicrobials

The first report of Ce use in the clinic dates from 1976 when William Monafo suggested that antimicrobial properties of the rare earth metals could benefit severely burned patients [40]. Sixty patients were treated with Ce. The protocol included early excision of wounds, usually within 5 days of injury, and wound coverage was provided by autografts. Ce nitrate treatment was well tolerated and yielded wounds ready to accept autografts. Additionally, there was a 50% reduction in mortality rate against predicted death rates. The bacterial profile also was different, with significant decreases in Gram-negative colonisation when cerium was used. Monafo also reported the treatment of eight burn patients with CN-SSD, which resulted in negative wound cultures twice that of patients treated with Ce alone. Overall, there was 50% reduction in mortality compared to predicted death rates from probit (a unit of measurement of statistical probability based on deviations from the mean of a normal distribution) tables [131] after the use of CN-SSD. It was presumed that this effect was a result of the antimicrobial effect of Ce; therefore, other studies attempted to establish its mechanism of action. Saffer et al. [132] studied the effect of Ce as a solution or cream added to a broth containing *P. aeruginosa* and found little antipseudomonal activity when added alone and only minimal synergism when used with SSD. Further tests using a model of *Pseudomonas* cultures inoculated into a wound on the dorsum of laboratory animals showed no antibacterial effect of cerium and reduced the efficacy of SSD.

Since then, further trials have generally supported the use of Ce, as improvements in overall outcomes have been demonstrated. One burns unit achieved a 39% reduction in mortality when compared to historical data [133]. Patients suffering from burns covering over 50 per cent of the body surface area were treated by topical application of a cream containing cerium nitrate (0.05 M) and silver sulphadiazine (0.03 M) (CN + SSD). Another study reported a 59% reduction in mortality against predicted death rates. This was accompanied by a fall in the level of wound colonisation and septic complications [134]. One randomised trial comparing CN-SSD against SSD did report higher mortality and levels of bacterial colonisation with Ce treatment; however, in this study, despite randomisation there was a higher number of older and more severely burned patients in the Ce-treated group [135].

A more recent randomised trial [51], which involved 60 patients and compared SSD and CN-SSD treatment, showed that re-epithelisation of a partial thickness wound occurred about 8 days earlier when treated with CN-SSD. A similar reduction in re-epithelialisation time was shown by Winter [136] when wounds were kept moist compared to those which were allowed to dry out, suggesting another possible mechanism for improved outcomes after Ce treatment. Excised full thickness wounds were ready to receive grafts 11 days earlier than SSD-treated patients. The average hospital stay was reduced by 7 days in the Ce group but there was no statistical difference in mortality between groups.

Historically, patients were bathed in a solution of CN or had gauzes soaked in CN applied to their wounds, but currently CN is usually applied as a cream that combines 2.2% CN with 1% silver sulfadiazine giving final concentrations of 0.05 M and 0.029 M respectively. The cream is available on the market as Flammacerium^®^ (Alliance Pharma PLC, Chippenham, UK) or Dermacerium (Silvestre Laboratories, Rio de Janeiro, Brasil). Flammacerium is used in western European countries, such as Belgium, the Netherlands, France and Spain and in the UK, as a topical treatment used in the treatment of burns. In the case of the UK, because of historical constraints over licensing, it is only available on a named patient basis. In 1999 it received orphan drug status in the United States, as it is not yet Food and Drug Administration-approved. In 2010 Signe-Picard carried out a retrospective study to determine the effectiveness of Flammacérium, in the stabilisation of necrosis in non-healing wounds and found it to be effective [137]. In a clinical trial aiming to assess the effectiveness of Flammacerium on ischemic necrosis wounds of the lower limb as an alternative to amputation for a period of 12 weeks, Vitse et al. found that the treatment was effective in the standardized care of ischemic necrotic wounds of the lower extremity [138]. A survey of British burns surgeons revealed that Flammacerium is the treatment of choice when immediate excision and closure is not practicable [139]. Its desirable properties include reduction of the bacterial colonisation of the burn wound and inflammatory response, and providing a more manageable eschar. It is in use in the majority of British burns units and, whilst most appear to have systems in place to allow adequate supplies of the product, most would prefer to see it fully licensed within the UK.

Apart from direct antibacterial effects, immunomodulatory properties have been recognised as a major mechanism by which cerium helps prevent sepsis in burn patients. In mice, cerium nitrate protects from post burn immunosuppression, reduces alterations of the splenic helper to suppressor lymphocyte ratio [140], and improves survival following septic challenge [141]. In humans, it seems to preserve normal T-cell functions such as production of interleukin 2 (IL-2) and IL-2 receptor expression [142]. Interleukin-2 acts as a growth factor for all T-lymphocyte subpopulations, activates natural killer cells, B-cell antibody production and induces other cytokines such as IL-1, TNF-α and TNF-β and interferon-γ [143]. It is a central regulator of the immune response and acts as a marker of T-cell-mediated immunity.

### 4.1. Effects in the Eschar

Since the earliest reports of use, the beneficial effects of Ce on the burn eschar (a dry layer of dead tissue found in a full-thickness wound following a burn or an infectious disease on the skin) has been a universal finding [43]. Topical application of Ce results in a firm eschar with an impenetrable leather-like appearance and greenish yellow discolouration. This eschar is firmly attached to the wound beneath in contrast to the soft macerated eschar generated by silver sulfadiazine treatment. The physical hardening of the eschar is beneficial in burn care, as soak-through and dressing changes are minimised and less labour intensive [144], and excision in most cases easier [135,145].

Boeckx et al. [9] compared the effects of SSD with and without cerium on the eschar. He studied twenty-two patients with deep dermal burns (TBSA range 10–77%) over a period of 8 months by selecting two site-matched areas treated either with SSD or CN-SSD. Punch biopsies were taken daily for histological examinations. On routine haematoxylin and eosin staining, the Ce-treated specimens showed a thin eosinophilic surface layer and a marked basophilic banding at the junction of the papillary and reticular dermis. The banding increased in width and intensity from the first post-burn day onwards. These changes were not present in the SSD group, which did, however, have a marked inflammatory infiltrate and evidence of early wound healing, which was absent in Ce treated specimens. Subsequent biopsies, taken up to 10 weeks after burn injury, showed no signs of wound healing beneath intact cerium crusts. Further examination showed deposits, that were present from the first post-burn day, of insoluble pyrophosphate and carbonate salts, as well as calcium deposits within upper dermal layers. All these deposits were located within the upper part of 1 mm biopsies and present only in the Ce-treated group. Calcium was not found in the absence of Ce in any specimen. The authors suggested that Ce might bind tissue pyrophosphate, removing the inhibition provided to local calcium deposition, similar to pyrophosphate-calcium interaction with cancellous or cortical bone. Such phenomena are not observed in case of cerium interaction with superficial burns where basal membranes remained intact, which suggests the importance of dermal collagen, which may act as a nidus for cerium pyrophosphate and subsequent cerium crystallisation.

These histological observations provide some, but not a full explanation, of the beneficial effects of cerium on mortality. It could be that the impenetrable crust prevents bacterial colonisation of the burned wound, reducing sepsis and improving the outcome. Alternatively, it could act as a barrier to the egress of burn toxins into the systemic circulation. It was shown experimentally that cerium binds to LPC and its precursors and denatures the toxic components [146]. Whatever mechanism is responsible, and it is likely to be a combination of both, the effects on the eschar are remarkable: it firmly adheres to the wound for many weeks with a minimal incidence of sub-eschar infection [45]. Eschars have been left in situ for 6 [133], 12 [9] and 14 weeks [45] without detriment. There is little tendency towards spontaneous eschar separation, and when finally excised, the wound underneath is healthy enough to accept a skin graft. The reported graft takes are 90%. There appears to be inhibition of granulation and contraction of full thickness wounds beneath the Ce eschar [134]. There is a report of slightly delayed re-epithelisation of superficial wounds for which Ce is probably not indicated [145].

### 4.2. Other Nanocomposites That Contain Cerium and Their Applications

Bifunctional components with both antibacterial and osteogenic-promoting properties could be incorporated into bone implants or bone regeneration materials for multi-functionalization. These kinds of bifunctional components could be synthesized by conjugating osteogenic and antimicrobial molecules together, such as copper, magnesium, cobalt and zinc, strontium, gallium, tantalum, and cerium [147]. In bone-related materials, the high content of Ga may induce cytotoxicity, which could be mitigated by co-doping with cerium. Cerium can dissociate the cell membrane of bacteria and shows low cytotoxicity [148]. Therefore, Ce^3+^ ions could enhance the antibacterial efficiency of bone substitute [149]. When applied with other bioactive ions, Ce^3+^ could extend the function of antimicrobial bone implants, especially hydroxyapatite-based materials. Ce^3+^/Sr^2+^ dual-substituted nano-hydroxyapatites exhibited better antibacterial and biocompatible activities than those of single-substituted nano-hydroxyapatites [150]. The cerium ion enhances the antimicrobial activity of Mg^2+^- and Sr^2+^-substituted hydroxyapatite [151]. Besides Ce^3+^, ceria also has outstanding antibacterial and anti-inflammatory properties. Nanostructured ceria-modified titanium implants possessed antibacterial property for peri-implantitis prevention, and the ROS-scavenging ability also relieved inflammatory responses [77]. Hammounda et al. [152] studied the biocompatibility of different scaffolds based on Ce-doped nanobioactive glass, collagen, and chitosan using the first passage of rabbit bone marrow mesenchymal stem cells (BM-MSCs) directed to osteogenic lineage by direct and indirect approaches. One percentage of glass filler was used (30 wt. %) in the scaffold, while the percentage of CeO_2_ in the glass ranged from 0 to 10 mol. %. The results showed that at 24 h after direct contact with the composite scaffold, all scaffolds showed proliferation of cells >50% and increased cell density on day 7. The scaffold with the highest percentage of CeO_2_ in bioactive glass nanoparticles (sample CL/CH/C10) showed the lowest inhibition of cell proliferation (<25%) at day 7. Moreover, an indirect cell viability test showed that all extracts from the four composite scaffolds did not demonstrate a toxic effect on the cells (inhibition value < 25%). Therefore, they concluded that the addition of CeO_2_ to the glass composition improved the biocompatibility of the composite scaffold for the proliferation of rabbit bone marrow mesenchymal stem cells directed to osteogenic lineage.

Several nanocomposites with antibacterial properties and potential applications as wound dressing are being developed. As an example, Zamani et al. [153] studied the effect of gelatin-polycaprolactone nanofibers containing CNPs on *P. aeruginosa* by assessing the minimum inhibitory and bactericidal concentrations of the nanoparticles in an ATCC reference strain and a clinical isolate strain, the expression of the genes shv, kpc, and imp, to determine whether exposure to the nanocomposites might change the expression of antibiotic resistance and cytotoxicity of the CNPs on the fibroblast, using flow cytometry. They found that minimum bactericidal concentrations for the ATCC and the clinical isolate of 50 µg/mL and 200 µg/mL were measured, respectively, when the CNPs were used. In the case of the scaffold containing cerium, the bactericidal effect was 50 µg/mL and 100 µg/mL for the ATCC and clinical isolate, respectively. Interestingly, exposure to the scaffold with cerium significantly decreased the expression of the genes shv, kpc, and imp. The authors concluded that concentration of CNPs and cerium–enriched scaffolds higher than 50 μg/mL can be used to inhibit the growth of *P. aeruginosa*. The fact that the scaffold containing cerium significantly reduced the expression of resistance genes means it has the potential to be used for medical applications such as wound dressings. Cao et al. [84] investigated cerium nanoparticle (CeNP)-loaded polyvinyl alcohol (PVA) nanogels and their application as wound bandages. The CeNP nanogel (Ce-nGel) was fabricated by the fructose-mediated reduction of cerium oxide solutions within the PVA matrix. The nanogel particle sizes were evaluated by transmission electron microscopy and determined to range from ∼10 to 50 nm. Additionally, glycerol was added to the Ce-nGels, and the resulting compositions (Ce-nGel-Glu) were coated on cotton fabrics to generate the wound bandaging composite. The cumulative drug release profile of the cerium from the bandage was found to be ∼38% of the total loading after two days. Antibacterial efficacy was reported for Gram-positive and Gram-negative microorganisms. Rapid wound healing in mouse models after 24 days Ce-nGel-Glu-treatment with less damage in comparison to the untreated wounds was reported, which led to the conclusion that Ce-nGel-Glu-based bandaging materials could be potential candidates for wound healing applications in the future. Appu et al. [154] developed nanocomposites of chitosan-coated cerium oxide (CS/CeO_2_ NCs) derived from aqueous extracts of tea using green chemistry. This novel polymer demonstrated excellent antibacterial and antifungal efficacy against foodborne pathogens such as *Escherichia coli*, *Staphylococcus aureus*, and *Botrytis cinerea,* with zones of inhibition of 13.5 ± 0.2 and 11.7 ± 0.2 mm, respectively. The results elucidated the potential of biosynthesized CS/CeO_2_ NCs to be utilized as potent antimicrobial agents in the food and agriculture industries. Another approach was proposed by Sadeghi et al. [155] by synthesized magnetite modified by Cr and co-modified by Cr and Ce, along with reduced graphene oxide (rGO)-based nanocomposites via facile hydrothermal and co-precipitation methods. The rGO-based samples showed proper magnetic behavior, high porosity, and vast specific surface areas. The high specific surface area provided more adsorptive active sites with higher potentials for the decomposition of *Staphylococcus aureus* (*S. aureus*) cells. The antibacterial performance of the samples against *S. aureus* was evaluated at 50 and 100 μg mL^−1^ through the colony-forming unit (CFU) method, and the minimum inhibitory concentration (MIC), and minimum bactericidal concentration (MBC) values were subsequently determined. As per results, not only chromium cations could effectively damage the DNA of bacteria, but also the antibacterial efficacy was further enhanced by co-doping of cerium and integration with rGO nanosheets. The antibacterial results were confirmed through changes observed in the morphology and topology of the bacteria before and after the treatment, using SEM and AFM analyses. Ultimately, the plausible *S. aureus* inactivation mechanism of the samples was disclosed. Naseri et al. [156] incorporated cerium oxide (CeO_2_) nanoparticles into poly(ε-caprolactone) to improve gelatin films as a potential wound dressing material. They electrospun PCL/gelatin (1:1 (*w*/*w*)) solution containing 1.5, 3, and 6% (*w*/*v*) of CeO_2_ nanoparticles to prepare the wound dressings. The morphology, contact angle, water absorption ability, tensile strength, water vapour transmission rate, and cellular reaction were evaluated, and the highest cell proliferation occurred with L929 cells in the PCL/gelatin film containing 1.5% (*w*/*v*) CeO_2_ nanoparticles. For in vivo analysis of the full-thickness of excisional wounds in Wistar rats, a film incorporating 1.5% CeO_2_ nanoparticles was used as the ideal dressing. After 2 weeks, the wound dressing containing CeO_2_ nanoparticles contributed to a substantial closure of almost 100% relative to sterile gauze, which showed approximately 63% wound closure. The findings presented evidence to support the potential applications for effective wound care with the CeO_2_ nanoparticle-containing dressing. Bharathi et al. [157] applied cerium oxide (CeO_2_) and peppermint oil (PM oil) on electrospun polyethylene oxide (PEO)/graphene oxide (GO) polymeric mats and demonstrated sustained antibacterial properties against *E. coli* and *S. aureus* as assessed by disc agar diffusion. An MTT assay showed that after incorporating CeO_2_ and PM oil, the nanofibrous mat revealed low cytotoxicity against L929 fibroblast cells. The nanofibrous composite mat’s wound healing activity could be increased because of the existence of active functional groups in PM oil, and the dual oxidation state of CeO_2_. Histology findings showed that, by facilitating wound contraction, improved collagen deposition, and re-epithelialization, the composite nanofibrous mat exhibited a fast-healing mechanism. Wounds treated with the prepared nanofibrous mats showed better wound healing compared to the control. These antibacterial electrospun nanofibrous mats could be used in biomedical applications for next-generation wound dressing materials. Kalaycıoglu et al. [158] synthesised CeO_2_ NPs that were developed by a green approach using *Zingiber officinale* extract to reduce the toxicity of the compounds in their synthesis. In the PVA/chitosan/CeO_2_ NPs hydrogel with 0 to 1% (wt), 5 nm cerium oxide nanoparticles were synthesized by a freeze–thaw process. Then, the antibacterial activity of PVA/CS hydrogels containing 0.5 and 1% CeO_2_ NPs was studied by quantifying the survival of bacteria (*Escherichia coli* and *Staphylococcus aureus*) which were in contact for 1, 2 and 3 h at 35 °C. The results showed that hydrogels containing 0.5% CeO_2_ NPs had good antibacterial activity after 12 h, and, compared with the control group, human dermal fibroblast cell viabilities existed for up to 5 days (more than 90%). Cerium nanoparticles that incorporate chitosan/PVA hydrogels may be a promising candidate as a wound dressing agent because they can effectively reduce wound infections without the use of antibiotics. Kannan et al. [159] developed an anti-Leishmania nano-drug using ultra-small functional maghemite (γ-Fe_2_O_3_) nanoparticles (NPs), which were surface-doped by [CeLn]^3^/^4+^, to enable effective binding of the polycationic polyethylenebyimine (PEI) polymer by coordinative chemistry. This resulting nano-drug is cytolytic in vitro to both *Trypanosoma brucei* parasites, the causative agent of sleeping sickness, as well as to three *Leishmania* species. The nano-drug, termed “Nano-Leish-IL” for topical treatment of cutaneous leishmaniasis (CL), induces the rupture of the single lysosome present in these parasites attributed to the PEI, leading to cytolysis and elimination of *L. major* infection in mice.

## 5. Controversies and Conclusions

With the global challenge of AMR, there is no doubt that we must improve antibiotic stewardship. However, whether nanoceria can synergistically enhance the antibacterial properties of antibiotics is controversial. Bellio [111] proposes that nanoceria could act as antibiotic adjuvant to increase the effectiveness of antimicrobials. Nanoceria increases bacterial outer membrane permeability, allowing the entrance of the antibiotics to increase their antimicrobial activity against MDR pathogens. However, in another study, it was found that when using nanoceria and ciprofloxacin together, the antimicrobial effect of ciprofloxacin could be dramatically reduced, by preventing its absorption into the bacterial cell or interfering with its interaction with bacterial DNA inside of cell [160]. When NaCe(MoO_4_)_2_ was combined with different antibiotics against bacteria, it showed either a synergistical or antagonistic effect [161]; therefore, more studies are needed to understand the relationship between nanoceria and antibiotics and their combined effect on bacteria.

The rare earth elements do not penetrate living mammalian membranes; therefore, adverse effects are infrequent. Hirakawa [57] identified a significant amount of silver in both kidneys and liver, but only a trace of Ce in hepatic tissue and none at all in the kidneys. Two early clinical studies [40,49] reported occasional cases of transient methoglobinaemia following Ce nitrate therapy. It is likely this was due to bacterial reductions of nitrate in the wounds, in a similar manner to that documented with silver nitrate use [162]. Another commonly reported adverse effect in clinical practice is a stinging sensation after application in up to 87% patients in one series, which can be easily controlled with oral analgesics [44].

Despite showing various promising biomedical applications, nanoceria also show some toxic effects, which are currently a significant concern. The toxicity of nanoceria depends on various factors, such as particle size, preparation method, cell type, dose/concentration, exposure time, and exposure route [163]. Aalapati et al. [164] studied the toxic effects and bioaccumulation of nanoceria in CD1 mice and observed that nasal inhalation resulted in pulmonary and extra pulmonary toxicity. Wu at al. [165] studied size–dependent toxicity of nanoceria on mice after repeated intranasal instillation. Two different nanoceria (7 and 25 nm) were used in their experiment to study the toxic effects of nanoceria in the lung, liver, spleen, kidney and brain in mice. They observed size-dependent pulmonary damage for nanoceria. On the other hand, both sizes of nanoceria showed similar systemic toxicity on other organs. In 2014, Kumari et al. [166] analysed the genotoxicity by administering repeated oral doses of nanoceria and cerium oxide microparticles in Wistar rats. They observed that nanoceria exhibited toxic effects without any severe distress symptoms and mortality at medium and high doses. They observed that long-term exposure of nanoceria at higher concentrations caused genetic damage (DNA damage in peripheral blood leukocytes and liver), histological damage (alterations in liver, spleen and brain) and biochemical alterations (alterations in lactate dehydrogenase and alkaline phosphatase activity in serum and reduction in glutathione content in liver, kidney and brain) in rats. Nanoceria also showed immunotoxicological effects by interfering with the immune system. Ranjbar et al. [167] studied the dose-dependent effect of nanoceria on rat liver by giving rats intraperitoneal injections. In 2018, Gagnon et al. [168] reported that exposure of nanoceria in natural water caused immunotoxicity in rainbow trout. They found that when the fish were exposed the nanoceria accumulated in their gills. The accumulation was more pronounced in ‘green water’, with higher pH, higher conductivity and containing less total organic carbon as compared to ‘brown’ water.

The antibacterial and immunomodulatory properties of cerium have been known for over a century. In this review, several medical applications have been addressed for cerium compounds based upon the antiseptic and immunomodulatory properties of Ce. All these applications are still controversial, except for its topical use for the treatment of burn wounds. Studies addressing the antimicrobial effects of Ce have consistently produced contradictory results. A lack of standardization between the studies, for both the bacterial species used and concentrations and formulations of cerium tested, makes them difficult to compare. Researchers have made some progress in the development of Ce-based antibacterial agents and biomedical materials, but far from enough. In order to progress the translational potential of cerium and cerium oxide-related antibacterial materials, better standardisation, more systematic studies, and observation of long-term effects should be conducted to improve understanding of the cytotoxicity and mechanisms of new cerium- and cerium oxide-based materials.

## Figures and Tables

**Table 1 molecules-27-02678-t001:** Timeline of cerium discovery and its antimicrobial applications.

Date		Ref.
1787	Arrhenius identifies ‘ytterbite’.	
1803	Cerium discovered by Berzelius and Hisinger.	
1894	Pokorny identifies cerium components much more toxic to bacteria then algae.	
1897	Studies by Drosseback on bacteriostatic activity of cerium components.	[35]
1912	Toxicity study of lanthaniun sulfate on tubercle bacillus by Froulin.	[53]
1915	Bohn testing range of Ce^3+^ solution on wide range of wounds	[36]
1936	Gould reports effect of cerium on enzymatic activity.	[54]
1947	First systematic analysis of antibacterial properties of cerium by Burkers	[37]
1976	First clinical study of antimicrobial properties cerium Nitrate on burns by Monafo.	[40]
1977	Introduction of silver sulfadiazine as topical antimicrobial agents by Fox.	[43]
1977	Combined antimicrobial therapy of burns using cerium nitrate and silver sulfadiazine by Fox.	[45]
1977	Combined therapy of cerium nitrate-silver sulfadiazine cream as a topical antiseptic agent for both major and minor burn wounds in children by Monafo	[55]
1979	First randomised study of cerium nitrate-silver sulfadiazine cream in the treatment of burns by Helvig.	[56]
1983	Determination of liver and kidney toxicity of silver and cerium nitrate from a severely burned infant by Hirakawa.	[57]
1985	Introduction of cerium-Flamazine cream for burns treatment by Boeckx.	[58]
2004	Silver sulfadiazine and cerium nitrate used for treatment of oxacillin- and mupirocin-resistant *Staphylococcus aureus* hospital strains by Schuenck.	[59]
2006	Report of methemoglobinemia after Flammacerium (cerium nitrate + silver siladiazine cream) treatment by Attof.	[60]
2006	Synthesis of cerium oxide nanoparticles by Garidi.	[61]
2007	CeO_2_ nanoparticles synthesis using egg white by Maensiri.	[62]
2009	Report of topical application of cerium nitrate preventing burn oedema in rats by Kremer.	[63]
2010	First report of high chloraemia in patients with deep third-degree burns treated with Flammacerium by Chianea.	[64]
2010	Introduction of cerium dioxide nanoparticles as antiviral agent by Zholobak.	[65]
2010	Study on effects of engineered cerium oxide nanoparticles on bacterial growth and viability by Pelletiner.	[66]
2012	Study on antibacterial activity of polymer coated cerium oxide nanoparticles by Shah.	[67]
2013	In vivo study on antibiofilm effect of cerium nitrate against *C. albicans* by Cobrado.	[68]
2014	Synthesis of gold-supported cerium oxide nanoparticles for antibacterial applications by Babu.	[69]
2015	Study by Selvaraj et al. study indicating that CeO_2_ nanoparticles may be useful for the treatment of sepsis.	[70]
2015	In vitro study on antifungal activity and in vivo antibiofilm activity of cerium nitrate against Candida species by Silva-Dias.	[71]
2015	Green Synthesis of cerium oxide nanoparticles using *Gloriosa superba* L. leaf extract with antibacterial properties by Arumugam.	[72]
2016	Synthesis of bimodal, ZnO:CeO_2_:nanocellulose:polyaniline bionanocomposite with capacity to absorb dissolved Arsenic along with a noticeable antibacterial activity by Nath.	[73]
2017	Study on production size controlled ultrafine CeO_2_ nanoparticles with antibacterial activity using microwave by Al-Shawafi.	[74]
2017	Fabrication of biopolymer-based silver-cerium-chitosan nanocomposite wound dressing with wound healing and antimicrobial properties by Es-Haghi.	[75]
2017	Synthesis of heterostructured cerium oxide/yttrium oxide nanocomposite with antibacterial properties in UV light induced photocatalytic degradation and catalytic reduction by Magdalane.	[76]
2019	Study on antibacterial and anti-inflammatory capabilities of surface-treated titanium implants via nanostructured ceria by Li.	[77]
2019	Study on antimicrobial activity of plasma-sprayed cerium oxide-incorporated calcium silicate coating in dental implants by Qi.	[78]
2019	Engineering the Bioactivity of Flame-Made Ceria and Ceria/Bioglass Hybrid Nanoparticles by Matter.	[79]
2020	Synthesis of silver-cerium titanate nanotubes for antibacterial applications by Sales.	[80]
2021	Study on synergistic antimicrobial potential of nitric oxide (NO) donor molecule and cerium oxide nanoparticle (CNP) by Estes.	[81]
2021	Synthesis of molybdenum disulphide-ceria (MoS_2_-CeO_2_) nanocomposite with photo-thermal therapy (PTT) antibacterial capability by Ma.	[82]
2021	Antibacterial study of Ag/cellulose-doped CeO_2_ quantum dots by Ikram.	[83]
2021	Study on antibacterial and wound-healing properties of cerium oxide nanoparticle-loaded polyvinyl alcohol nanogels bandages by Cao.	[84]

**Table 2 molecules-27-02678-t002:** Summary of antibacterial activity of nanoceria.

Year	Particle Size/Morphology	Type of Bacteria	Concentration	Findings	References
2017	3–4 nm/spherical	*Pseudomonas aeruginosa* and *Staphylococcus epidermidis*	250 and 500 µg/mL	Cerium oxide nanoparticles exhibited a perfect antibacterial activity against the bacteria at basic pH as compared to acidic pH values due to a smaller size and positive surface charge at pH 9	[117]
2017	3.5–6.5 nm	*Escherichia coli*	N/A	Nanoceria significantly inhibited the growth of *E. coli.* The rates of bacterial growth inhibition were found to depend on the average sizes and concentration of the nanoceria	[113]
2015	5 nm/spherical	*Staphylococcus aureus*, *Streptococcus pneumonia*, *Escherichia coli*, *Pseudomonas aeruginosa*, *Proteus vulgaris*, *Klebsiella pneumoniae* and *Shigella dysenteriae*	Antimicrobial discs loaded with 100,000 µg CeO_2_ nanoparticles	Nanoceria showed a strong antibacterial activity and Gram-positive (G+) bacteria were relatively more susceptible to the NPs than Gram-negative (G−) bacteria. The toxicological behavior of CeO_2_ NPs was found due to the synthesized NPs with uneven ridges and oxygen defects in CeO_2_ NPs.	[72]
2014	5 nm	*Streptococcus mutans*	220 µg/mL	Nanoceria seemed to be a very effective antimicrobial agent against *Streptococcus mutans* probably by destroying cell walls as a result of reactive oxygen species production	[118]
2006	7 nm/ellipsoidal	*Escherichia coli*	0 to 730 µg/mL	Positively charged at neutral pH nanoparticles display a strong electrostatic attraction toward Gram-negative *E. coli* outer membranes resulting in cytotoxic effect	[95]
2011	7 nm and 25 nm/truncated octahedral rhombus or irregular	*Escherichia coli*	10, 100 and 200 µg/mL	Direct contact of CNPs with the surface of *E. coli* causes a rise in intracellular ROS level, which results in antibacterial activity. Due to agglomeration and negligible effect on membrane integrity, 7-CeO_2_ did not exhibit greater antibacterial activity than 25-CeO_2_.	[100]
2012	8–10 nm	*Escherichia coli*	4.3 µg/mL	Dextran-coated CeO_2_ are non-toxic or exert mild anti-bacterial activity to *E. coli.* The toxicity of CeO_2_ NPs depends on the physical and chemical environment; what is more, the cerium oxide nanoparticles can decrease the anti-bacterial activity exerted by magnesium and potassium salts.	[67]
2018	10 nm	*Escherichia coli* and *Klebsiella pneumoniae*	50–600 µg/mL	Inactive nanoceria can exert a synergistic action capable of enhancing the activity of β-lactam antibiotics. CeO_2_ NPs increases the effectiveness of antimicrobials and activity is compromised by drug resistance mechanisms.	[111]
2014	10–20 nm	*Escherichia coli*, *Klebsiella pneumoniae*, *Salmonella enterica*, *Staphylococcus aureus* and *Enterococcus faecalis*	5000, 250,000 and 500,000 µg/mL	Nanoceria-doped composite nanofibers have demonstrated effective toxicity against both the Gram-positive and Gram-negative bacterial strains by disrupting bacterial cell membranes leading to irreversible damage to the cell envelope, which eventually results in cell death	[119]
2020	Between 10 and 20 nm/spherical or quasi spherical	*Klebsiella pneumonia*, *Staphylococcus epidermidis*, *Bacillus subtilis*, *Pseudomonas aeruginosa* and *Escherichia coli*	250–4000 µg/mL	Nanoceria were able to inhibit the bacterial strains across the tested concentrations ranging from 4000 µg/mL to 250 μg/mL, except for *E. coli* and *P. aeruginosa* that appeared resistant to low doses of nanoceria. *B. subtilis* appeared as the most susceptible strain	[120]
2016	11 nm/spherical	*Staphylococcus aureus*, *Pseudomonas aeruginosa*, *Escherichia coli* and *Klebsiella pneumoniae*	1000–5000 µg/disc	Nanoceria exhibited antimicrobial activity. Moreover, they showed the inhibition of respective bacterial biofilm formation	[121]
2014	25 nm	*Escherichia coli*, *Salmonella typhimurium*, *Bacillus subtilis* and *Enterococcus faecalis*	4, 8 and 16 µg/mL	Bacterial toxicity leading to cell death resulted from the direct interaction between nanoceria and bacteria on CeO_2_ NPs embedded nanocomposite membrane	[115]
2014	25–30 nm/elliptically spherical	*Escherichia coli* and *Staphylococcus aureus*	N/A	Nanoceria, synthetized from *Acalypha indica* leaf extract inhibited bacterial growth by 90%. The antibacterial properties were concentration-dependent.	[122]
2012	25–50 nm	*Escherichia coli*	5000 µg/mL	After UV irradiation (2 h), metal-oxide NPs inhibited the growth of *E. coli* due to oxidative stress (superoxide radical, hydroxyl radical, and singlet oxygen generated by TiO_2_ nanoparticles and ZnO nanoparticles)	[106]
2016	27 nm/spherical	*Staphylococcus aureus*, *Streptococcus pyogenes*, *Pseudomonas aeruginosa*, *Klebsiella pneumonia.**C. albicans*, *F. oxysporum*, *A. niger* and *A. candidus*	200 µg/mL	Interaction of bacterial and fungal cells with CeO_2_-CdO nanocomposite causes cell death due to generation of reactive oxygen species	[123]
2017	40–100 nm/spherical, cubical and circular	*Corynebacterium diphtheria*, *Sarcina lutea*, *Escherichia coli*, *Proteus vulgaris*	5000–20,000 µg/disc	Gram-negative bacteria were more susceptible to nanoceria in comparison to Gram-positive bacteria	[124]
2015	42 nm/spherical	*Pseudomonas aeruginosa* and *Staphylococcus aureus*	10,000–20,000 µg/mL	Increased of zone of inhibition in correlation with increased concentration of nanoceria, but only in case of *P. aeruginosa* (G−)	[125]
2019	<50 nm	Biofilm originated from *Citrobacter* and *Pseudomonas* species	0.05–200 µg/mL	Nanoceria accelerate biofilm formation due to oxidative stress	[126]
2013	100 nm/octahedral or truncated octahedral	*Escherichia coli*	75–30,000 µg/mL	The interaction of nanoceria with non-ionic surfactants (Triton X-100, Polyvinyl Pyrrolidone (PVP) and Tween 80 with, 0.001% *v*/*v*) enhanced their antibacterial activity against *E. coli*	[127]
2008	140 nm	*Escherichia coli*	10,000 µg/mL	Illumination of *E. coli* in the presence of hollow ceria nanospheres coated with conductive polymers (polyaniline and polypyrrole) decreased bacteria concentration	[114]

## Data Availability

Not applicable.

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
