# Peer review of "The Use of Cerium Compounds as Antimicrobials for Biomedical Applications"

_molecules, 2022, doi:10.3390/molecules27092678_

Round 1

Reviewer 1 Report

The manuscript submitted by barker et al. describes the potential applications of cerium in different scenarios. However, the review needs to be improved.

Many papers have been published in the literature in 2021/2022. Therefore, the authors must include them and expand the review. 

Besides that, please address the following:

1. Consistently use the name of the element in small letters. Sometimes is annotated with capital "C."

2. Table 2  should be rearranged with significant changes. (a) To provide a fast comparison, please unify the concentrations in ug/mL and not mix values (for example, in percentages). (b) Instead of the concentrations, use the concentrations that showed the MICs. (c) I suggest arranging this table according to either the size or the morphology of the NPs, rather than the year. (d) The table should include more descriptive findings. For example, when mentioning surfactants, please add what surfactant. All these findings should be developed/explained after the table mentions these activities' rationale.

3. Classify the activities in subheadings: antimicrobial, wound healing, etc.

4. Also, include other nanocomposites that contain cerium and explain the add-ons of the inclusion of this element in the nanocomposite.

5. Add other potential applications with cerium using this nanocomposite (in #4).

Author Response

The manuscript submitted by barker et al. describes the potential applications of cerium in different scenarios. However, the review needs to be improved. Many papers have been published in the literature in 2021/2022. Therefore, the authors must include them and expand the review.

Authors’ comment: The review has been expanded in subsection 4b. Other nanocomposites that contain cerium and their applications have been included in this new section (see highlighted text in yellow in the submitted revised version)

Besides that, please address the following:

  1. Consistently use the name of the element in small letters. Sometimes is annotated with capital "C."

Authors’ comment: It has now been corrected.

  1.  Table 2  should be rearranged with significant changes.

(a) To provide a fast comparison, please unify the concentrations in ug/mL and not mix values (for example, in percentages).

Authors’ comment: It has now been done.

(b) Instead of the concentrations, use the concentrations that showed the MICs.

Authors’ comment: The concentrations have been changed.

(c) I suggest arranging this table according to either the size or the morphology of the NPs, rather than the year.

Authors’ comment: Table 2 has been – re-formatted (see changes highlighted in yellow in the revised version of the manuscript).

(d) The table should include more descriptive findings. For example, when mentioning surfactants, please add what surfactant. All these findings should be developed/explained after the table mentions these activities' rationale.

Authors comment: The purpose of the table was to act as a summary and the authors believe adding an extra paragraph after the table explaining the findings would unnecessarily lengthen the review. We do agree with the reviewer with the fact that more information was needed and therefore key additional descriptive comments have been now added.

  1. Classify the activities in subheadings: antimicrobial, wound healing, etc.

I am afraid the reviewer request is not clear to the authors as we already have divided the review into key subheadings preceded by an index.

  1. Also, include other nanocomposites that contain cerium and explain the add-ons of the inclusion of this element in the nanocomposite.

Authors’ comments: It has been commented in new subsection 4b (see text highlighted in yellow).

  1. Add other potential applications with cerium using this nanocomposite (in #4).

Authors’ comment: Also highlighted now in sub-section 4b.

Reviewer 2 Report

This review is well written, and it reviews the applications of cerium compounds comprehensively as antimicrobials. The history and usages of cerium and the antibacterial activity of nanoceria are organized well in the tables, providing a clear view to the readers. The controversy of using cerium compounds is also revealed. This article is highly recommended to be published in the Molecules.

Author Response

Reviewer 2 

No changes have been required by this reviewer and the authors wish to thank the reviewer for the positive and encouraging comments.